# Mechanical Properties of Thin-Ply Composites Based on Acoustic Emission Technology

**DOI:** 10.3390/ma14040913

**Published:** 2021-02-15

**Authors:** Kaidong Zheng, Dongfeng Cao, Haixiao Hu, Yundong Ji, Shuxin Li

**Affiliations:** 1State Key Laboratory of Materials Synthesis and Processing, Wuhan University of Technology, Wuhan 430070, China; 1049731700590@whut.edu.cn; 2Foshan Xianhu Laboratory of the Advanced Energy Science and Technology Guangdong Laboratory, Foshan 528000, China; 3School of Materials Science and Engineering, Wuhan University of Technology, Wuhan 430070, China; jiyundong@whut.edu.cn; 4Institute of Advanced Materials and Manufacturing Technology, Wuhan University of Technology, Wuhan 430070, China

**Keywords:** thin-ply laminate, acoustic emission, failure modes, characteristic frequency

## Abstract

Compared with standard-ply composites, thin-ply composites exhibit a superior mechanical performance under various operating conditions due to their positive size effects. Thin-ply laminate failure modes, including matrix initial damage (MID), matrix failure (MF), and fiber failure (FF), have been distinguished through a systematic acoustic emission (AE) signals analysis combined with scanning electron microscopy (SEM). First, the characteristic frequencies of various failure modes are identified based on unidirectional laminates ([90] _68_ and [0] _68_). Then, according to the identified frequencies corresponding to distinctive damage modes, four lay-up sequences (0_2_[[90_m_/0_m_]_ns_]0_2_, m = 1, 2, 4, 8, n × m = 16) with a constant total thickness are designed, and the effects of the number of identical plies in the laminate thickness on the damage evolution characteristics and the damage process under uniaxial tension loads are dynamically monitored. The obtained results indicate that the characteristic frequency ranges for MID, MF, and FF are identified as 0–85 kHz, 165–260 kHz, and 261–304 kHz, respectively. The thickness of identical plies has a significant effect on onset damage. With the decrease of the number of identical plies (i.e., m in the stacking sequences), the thin-ply laminates exhibit the initiation of damage suppression effects and crack propagation resistance.

## 1. Introduction

Thin-ply composites (i.e., composites made of plies with thicknesses as low as 20 μm) have recently received increasing attention. Not only do they broaden the laminate design space significantly, but they also provide enhanced strength and damage resistance due to positive size effects. The open research literature shows that the reduction in ply thickness can significantly improve both the initial and ultimate strengths of multidirectional laminates under tensile loads [1,2,3,4,5,6,7,8,9]. In open-hole tests of laminates, thin-ply composites can obtain a higher initial damage strength under a quasi-static tension and a longer fatigue life under tensile cyclic loads [2,6,7]. In bolted assemblies, the bearing strength is also improved, especially in hot and humid environments [6]. In the case of CAI (compression after impact), thin-ply composites provided an optimizing design method to achieve an increased residual strength, although the thinnest part did not necessarily provide optimized results [2,6].

The size strengthening mechanism is not an intrinsic property of a material and depends on the geometric constraints between adjacent plies. The co-constraint effects suppress damage accumulation and the growth of delamination in the thin-ply laminates [10,11,12,13,14]. The identification of damage initiation and evolution characteristics in the thin-ply laminates is fundamentally important in order to fully exploit the potential of thin-ply materials for structural applications.

The acoustic emission (AE) technique is a nondestructive testing method that is used to detect stress waves that result from the release of elastic energy due to local rapid unloading [15,16,17,18]. AE technology is also used to obtain dynamic information during the process of damage initiation and evolution, providing a basis for the safety evaluation of composite materials or construction [19,20,21,22,23,24]. In recent years, AE technology has been widely used to identify and distinguish different damage modes in the field of composite laminates. Some researchers have classified the damage of composite laminates during the loading process into several modes, such as matrix fracture, fiber breakage, fiber–matrix interface debonding, and delamination [25,26,27]. 

However, distinguishing the specific frequency ranges corresponding to various fracture modes based on testing samples with different geometric sizes did not take into consideration the fact that frequency is connected with size [19,25,28]. In addition, samples of a same geometrical size composed of a single material, such as a fiber bundle, do not account for friction between fibers, which changes the sample frequency [19,26,27,28].

Previous work on thin-ply composites focused primarily on onset strength determination and fracture mode discrimination. However, AE technology applications in thin-ply composite laminates are unspecific, and the relationship between fracture mode discrimination and the characteristic damage frequency is limited by the sample geometry [25]. In addition, in previous works, the characteristic damage frequencies were usually identified by a pure resin matrix or a resin matrix within a fiber tow, which is not accurate enough. In fact, the characteristic frequencies are dependent on the fiber volume fraction and the interaction between fibers. 

In this work, based on the AE technique, the characteristic frequencies of matrix initial damage (MID), matrix failure (MF), and fiber failure (FF) are identified by unidirectional laminates ([90] _68_ and [0] _68_), whose material and geometry sizes are the same as pending test samples. Four lay-up sequence (0_2_[[90_m_/0_m_]_ns_]0_2_, m = 1, 2, 4, 8, n × m = 16) laminates with a constant total thickness are designed. The effects of the number of identical plies in the laminate thickness on the damage evolution characteristics and damage process under uniaxial tension loads are dynamically studied. Optical and scanning electron microscopy (SEM) are also employed to help certify the fracture model. 

## 2. Materials and Methods

### 2.1. Test Specimen Preparation

Specimens were made from ultra-thin prepregs with a ply thickness of 0.03 mm by hand layup and autoclave curing. Glass-reinforced plastic sheets were adhered to both loading ends of a composite plate by rapidly curing epoxy adhesive. Twenty-four hours later, the assembly was divided into individual test specimens of around 25 mm in width and 250 mm in length with waterjet cutting. An ultrasonic C-scan was adopted to determine the effectiveness of the molded plate and test specimens. Six different layups, specified in Table 1, were designed. Groups E and F were used to calibrate the characteristic frequencies of MID, MF, and FF. The remaining four groups (A, B, C, and D) were designed to study the effects of the ply thickness on the initial damage and evolution mechanics. The material grade of the prepregs was HRC2-45%-A3-U-30gsm-1000 and was provided by Jiangsu Hengshen Co., Ltd., Danyang, China. The prepared specimens are exhibited in Figure 1.

### 2.2. Test Methods

According to ASTM3039, a series of tensile tests was conducted. The specimens were loaded on an electro-hydraulic universal machine (MTS Landmark 370.10, Minneapolis, MN, USA) with a displacement rate of 2 mm/min. Two AE sensors at a distance of 100 mm from each other were placed on the specimen to collect signals, as shown in Figure 2b. GT800 digital full waveform AE detection systems (Hunan Enti Technology Co., Ltd., Henan province, China) were used to demodulate the signals. Three strain gauges (among them, two on the front surface and one on the back surface, as shown in Figure 2b) were employed to monitor the strains and possible asymmetrical deformations. Simplified waveforms of standard AE signals (including duration, channel threshold, amplitude, counts, rise time, and released energy) are shown in Figure 3. The AE signal processing and analysis method of this work were based on characteristic parameter filtering, and the threshold value was dependent on the testing environment (noise, vibration of the testing machine, and so on). According to our testing environment and relevant literature [25,29,30], the channel threshold was defined as shown in Table 2. Other AE characteristic parameters are also listed in Table 2. The 2B pencil core was broken before each test to ensure the validity of the received signals.

Figure 3 shows that the area below the envelope is an AE energy signal reflecting the relative energy magnitude or strength of AE events. The energy calculation of the AE signal is processed by the root mean square (*V_rms_*) or mean square (*V_ms_*) of the voltage. The *V_ms_*, *V_rms_*, and relative energy (*E*) of a signal are defined as Equations (1) and (2), as follows [23]. According to the fundamental principles of acoustic emission, the change of *V_ms_* with time is the energy change rate of the AE signal, and the total energy E of the AE signal in the period from *t*1 to *t*2 can be expressed in Equation (3).
(1)$$V_{ms}=\frac{1}{\Delta T}\int_{0}^{\Delta t}V^{2}\left(t\right)dt$$
(2)$$V_{rms}=\sqrt{V_{ms}}$$
(3)$$E\propto \int_{t1}^{t2}{\left(V_{rms}\right)}^{2}dt=\int_{t1}^{t2}V_{ms}dt$$

In above equations, Δ*T* and *V*(*t*) are the average time and signal voltage varying with time, respectively, while *t*1 and *t*2 refer to the time at which the start and end points of a single waveform signal are above the threshold.

### 2.3. Microstructural Analysis

The size of the samples for the optical and SEM observation was approximately 4 mm × 4 mm × 2.6 mm. The sections of prepared specimens were polished with diamond grinding paste, then cleaned ultrasonically for a structural characterization inspection at three scales (macroscopic, mesoscopic, and microscopic scales, respectively). Figure 4a gives the macroscopic characteristics of group A observed with an optical microscope. Figure 4b,c shows the mesoscopic and microscopic morphology of group A by SEM. The results showed that the fiber distribution was relatively uniform, without obvious voids or manufacturer defects, and that multiple layers were laid at the same angle without obvious boundary phases. This type of multiscale, in-depth observation ensures proper AE damage detection.

## 3. Results

### 3.1. Damage Mode Identification

Unidirectional composite laminates (groups E and F) were designed to distinguish the characteristic frequencies of MID, MF, and FF, respectively. Compared with specimens such as the pure resin matrix and the resin matrix within a fiber tow [25], the unidirectional composite laminate was designed to take the geometry effects on the AE frequency into consideration. Pure resin matrices or fiber tow specimens with the same geometric size show enhanced friction between fibers when compared with standard specimens during the processing from the fiber relaxation state to tightening [27,28].

In this article, the specific damage models were classified as MID, MF, and FF. The classification shows that delamination damage was neglected and did not belong to the primary damage mode in the process of thin-ply composite tensile testing. Moreover, delamination is only a phenomenon related to the failure of the matrix and fiber/matrix interface.

Various damage modes were shown to correspond to characteristic frequencies of the uniaxial tensile testing of unidirectional laminates. Figure 5a presents the AE signals, such as the frequency and amplitude, from the unidirectional laminate designed as [90]_68_ during the tensile process, and six specimens were tested to ensure reliable results. The AE signal from group E showed two specific frequency ranges: 0–85 kHz and 240–260 kHz. The AE signals in both frequency ranges had lower amplitudes, but the one in the frequency range of 0–85 kHz appeared earlier. The dynamic recording of AE signals is shown in Figure 5a, where the signals presented at a lower frequency range were detected earlier than at a higher frequency range. 

According to previous research [25] and a general understanding of the processing of composite failures, there is a close correspondence between the characteristic frequency and damage model. To verify the tensile damage model for group E, SEM images of the failure section morphology with reference to 90° tensile damage are shown in Figure 6. Figure 6a,b presents failure section morphologies with various magnifications and focused regions. The detected AE signals were derived from matrix damage, or MID and MF, and matrix exfoliation is rare in these types of systems. Therefore, it can be concluded from the AE signals and SEM image analysis that the lower frequency range is related to MID and that the higher frequency range is connected to MF, respectively.

Fiber fracturing causes a decreased structural stiffness and bearing capacity directly in composite structures, indicating oncoming structural failure. Therefore, AE signals referring to fiber fracturing usually occur last with higher amplitudes, which is similar to the conventional understanding. As shown in Figure 5b, the AE signals coming from group F were relatively complicated, and higher signal amplitudes were obtained from group F when compared to group E. Moreover, the AE experiment of group F has excellent repeatability. Therefore, a conclusion can be drawn that the fiber aggregation fracture frequency should be in the range of 261–304 kHz. 

The complete failure mechanism of composite laminates includes MID, MF, FF, and fiber/matrix interface failure. However, the analysis and summary of previous research results show that the frequency range involved in fiber/matrix interfacial failure always exists between MID and FF. To ensure an enhanced validity for this analysis, only the matrix and fiber damage modes were considered in this article’s scope. A summary of our AE specific frequency signals is listed in Table 3.

### 3.2. Characteristics of Acoustic Emission Signals

#### 3.2.1. Acoustic Amplitude and Accumulation Release Energy–Time Relationship

The AE accumulation energy signal is proportional to the area of the AE waveform, as shown in Figure 3. The energy unit V^2^s (see Equation (3)) was adopted to avoid voltage rectification and the effective value calculation.

Figure 7 shows the relationship of the AE amplitude and accumulation energy–time, where the instantaneous energy release events accompany the generation of higher amplitude AE signals, and the energy–time curves feature a multistage staircase consistent with Ref [23]. The fact that the literature presented a multistage staircase in the same thickness layer composite but with various matrix and fiber types is particularly worth mentioning [23]. This multistage staircase phenomenon reflects the essence of energy release events, such as fiber failure and unstable matrix crack propagation. As Figure 7A,B,D shows, the initial jump of accumulation energy corresponds to an increase in time but excludes Figure 7C. An explanation for group C’s layup blocked stacking designs with an unlike response is that this specific thickness (0.12 mm) provides the best protective effect on the fiber.

AE monitoring provides evidence related to laminate onset damages and damage evolution. The effective AE signal corresponding to onset damage was confirmed by analytical methods of AE signals based on characteristic parameters. The decreased tendency towards the time of the first effective onset damages signal is shown in Figure 7. Therefore, thin-ply laminates with an excellent matrix damage suppression and resistance were validated using uniaxial tensile testing with the assistance of AE detection technology.

#### 3.2.2. Acoustic Emission Accumulation Counts Analysis

An effective reflection method for the intensity and frequency of damage events is the accumulation count of events. The accumulation counts of the three damage modes (FF, MID, MF) that this paper is concerned with were distinguished based on the corresponding characteristic frequencies of damages adopted to explain the dynamic evolution process of various damages to the uniaxially tensile tested samples. 

AE accumulation counts are the accumulation of burst event counts, and the sharp increase of the accumulation curve belongs to the resulting damage accumulation over a short time. This phenomenon of accumulation count can directly reflect the dynamic evolution of various damage modes during the experiment.

Figure 8 gives the accumulated counts of the dynamic development process of three separate damage modes during the loading period. The whole perspective, the sum of accumulated counts of three diverse damage modes, increases from Figure 8a to Figure 8d. Figure 8 shows that the ratio of accumulation event counts of FF to total damage counts is relatively low and that the accumulation count curves of MID and MF present an intertwined state after damage occurred. This intertwined phenomenon is particularly evident in Figure 8a,b. However, Figure 8c,d does not show this phenomenon, but there was an obvious and increased gap between the curves of MID and MF. These phenomena reflect that the thickness of the blocked stacking laminate composite increases and the number of MF or matrix unstable propagation decreases, implying that the number of intralaminate transverse penetration cracks decreases. The damage evolution behavior of various thickness composites corresponds well to the research given in Ref. [31].

### 3.3. Strength Analysis

AE technology was used to monitor the initial damage strength of thin-ply composite materials. In this paper, the initial damage strength is defined as the stress corresponding to the first effective AE signal appearing in the tensile testing, and the ultimate failure strength belongs to the maximum stress sustained by the structure during the loading period. In addition, a special strength based on the stress corresponding to the first occurrence of the fiber fracture AE signal is proposed in order to calibrate the initiation of obvious damage to the structure.

No distinct local unloading phenomenon emergency in the stress–strain curve is shown before the structural failure, and all curves present a brittle fracture mode to the process of tensile testing, as shown in Figure 9a. The concerned strengths extracted in the tensile experiment were plotted as shown in Figure 9b, and it was clearly observed that the initial damage strength and ultimate failure strength were in good agreement, related to the variation tendency, with the literature [3], namely, the enhanced strengths were obtained by reducing the continuous block stacking thickness. Although the initial damage strength is consistent with the results of other researchers [3,25], the overall initial damage strength is lower, which is possibly attributable to the inconsistency of the material types. This paper proposes a special damage strength, another strength calibration based on the fiber fracture of the AE signal. It aims to avoid the early failure signal of the local matrix caused by a stress concentration on the pore position of the material with pore defects, thus underestimating the residual strength of the structure. According to a previous conjecture, the layup design of group C showed a better protective effect on the fibers. Therefore, the trends of the special damage strength (except group C) with respect to the stacking thickness correspond well to the initial damage strength, as shown in Figure 9b.

### 3.4. Fracture Mode Analysis

Strength reflects the local characteristics of the material and is closely related to the damage mechanism of materials. To explore the relationship between the tensile strengths and fracture morphologies of the composite laminate, and to characterize the evolutionary behavior of damage, fracture sections were randomly selected from the four groups of failure specimens. Optical microscopy and scanning electron microscopy were embraced in order to depict the fracture morphologies, as shown in Figure 10 and Figure 11, respectively. 

The structural strength dominated by the fiber fracture depends on the utilization degree of the fiber performance and on the various fracture modes corresponding to different degrees. The accumulative damage of the failure modes can exert a maximum fiber strength that is contrasted with other modes, such as a bundle of fiber failure [32]. 

The results show that the fracture morphologies transfer from neat (Figure 10a,b) into a ladder (Figure 10c,d); namely, the fracture modes under tensile loading directly demonstrate a transition from an accumulative damage failure (Figure 10a,b) to a bundle of fiber failure (Figure 10c,d).

Figure 6 reveals that the interface between the matrix and fiber belongs to a strong interfacial action. Therefore, an inference can be proposed that the initial cracks begin in the transverse matrix, then extend to the interfaces between the matrix and fiber or the adjacent ply interfaces, and present a dramatic drop in the stress–strain curve dominated by the fiber failure. 

In groups A and B, the transverse propagation of the matrix cracks in the thin layer were restricted by adjacent plies and were difficult to propagate to the plies’ interface. The cracks at the matrix were primarily distributed in the form of multiple fine cracks, resulting in a neat fracture. However, the accumulation layer thickness is enough to weaken the boundary constraint from adjacent plies in groups C and D. Therefore, the fracture morphology shows a ladder shape, which is caused by transverse penetration cracks with a discrete distribution and propagation.

## 4. Conclusions

Compared with previous research methods, such as a pure resin matrix or resin matrix within a fiber tow, a more reliable method based on a unidirectional laminate was employed to identify the characteristic frequencies of various failure modes. Based on the identified characteristic frequencies, the mechanical properties and damage evolution of thin-ply laminate with a four stacking sequence were studied.

The main results can be summarized as follows:(1)The characteristic frequency ranges for MID, MF, and FF were identified as 0–85 kHz, 165–260 kHz, and 261–304 kHz, respectively. Based on the characteristic frequencies, it was proposed that the initial damage signal of the fiber could be regarded as the basis of an obvious damage initiation calibration, for the reason that there was a certain correlation with the determination of the first effective damage signal.(2)For (0_2_[[90_m_/0_m_]_ns_]0_2_) laminates, the thickness of identical plies has a significant effect on the onset damage. When the number of identical plies (i.e., m in the stacking sequence) was the minimum, the time of the first effective onset damages signal came last. This indicated that thin-ply composites could exhibit initiation damage suppression effects and crack propagation resistance.

This research work is mainly applied to the online damage monitoring of composite laminates, providing a new approach to the damage mode identification of composite laminates, and it puts forward another suggestion for the judgment of damage initiation. However, it is limited to the material system, where the matrix strength is obviously weaker than the interfacial strength, and out-of-plane damage monitoring such as delamination needs to supplemented by other experiments. 

## Figures and Tables

**Figure 1 materials-14-00913-f001:**
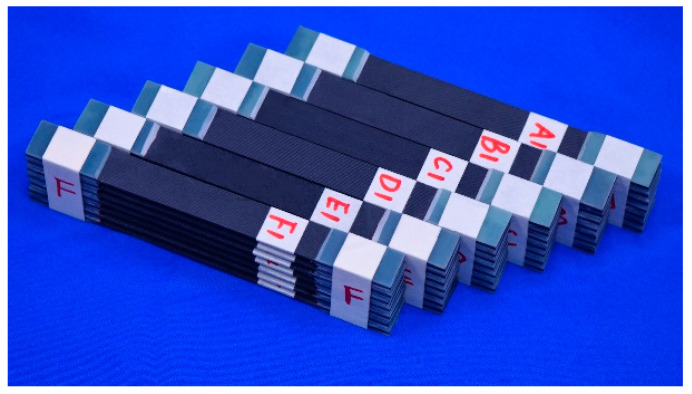
Image of the samples to be tested.

**Figure 2 materials-14-00913-f002:**
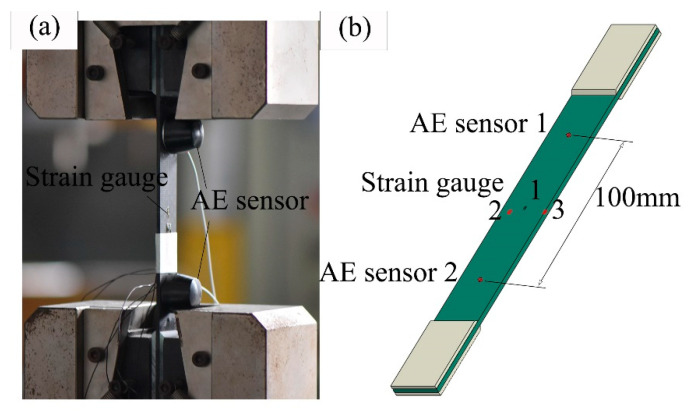
Tensile experiment with acoustic emission detection. (**a**) Fixture and (**b**) locations of the AE sensors and strain gauges.

**Figure 3 materials-14-00913-f003:**
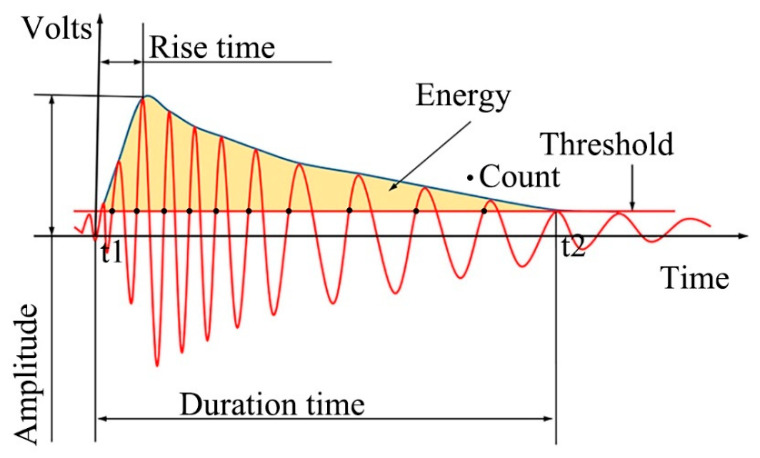
Simplified waveform of standard AE signals [23,25,29].

**Figure 4 materials-14-00913-f004:**
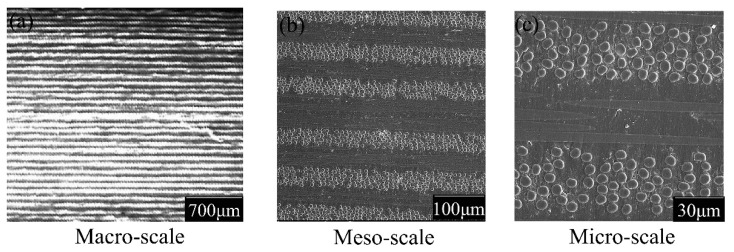
Sample A morphology before testing ((**a**) Macroscopic section morphology, (**b**) Mesoscopic section morphology, and (**c**) Microscopic section morphology).

**Figure 5 materials-14-00913-f005:**
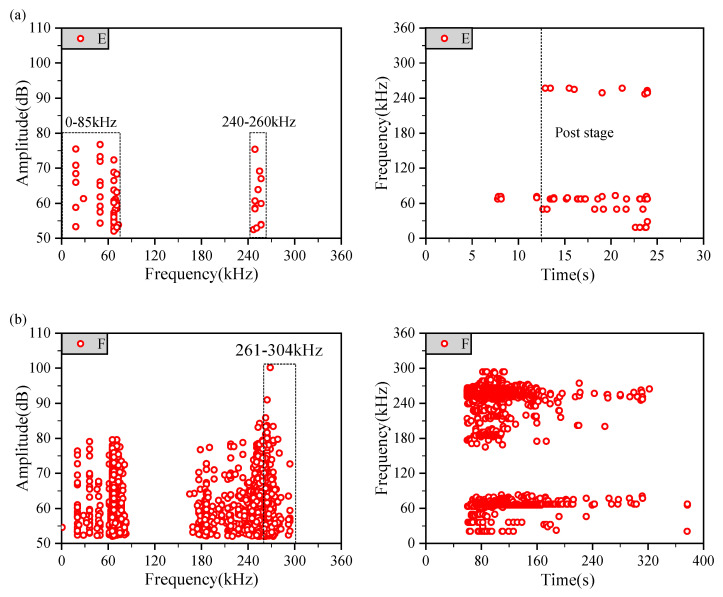
Amplitude–Frequency and Frequency–Time relationship graphs of AE ((**a**) Unidirectional laminate [90]_68_, (**b**) Unidirectional laminate [0]_68_).

**Figure 6 materials-14-00913-f006:**
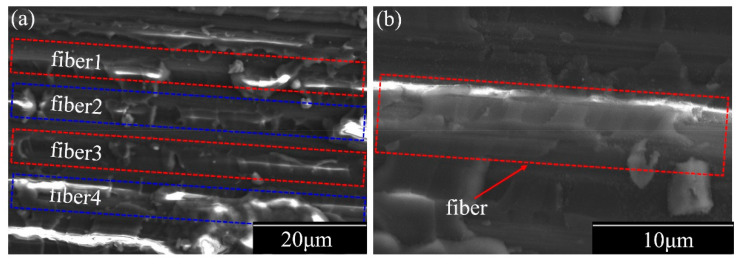
SEM images of the failure section of the 90° tensile damage morphology. (**a**) at the 20 μm scale, (**b**) at the 10 μm scale.

**Figure 7 materials-14-00913-f007:**
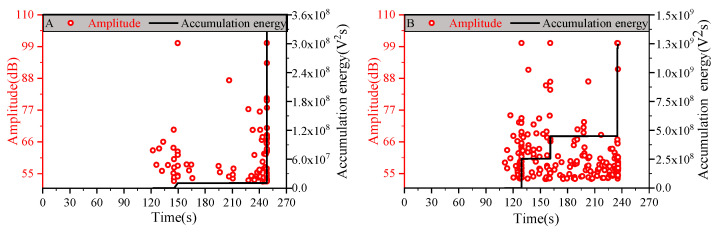
Acoustic amplitude and accumulation energy–time relationship graphs: (**A**) Group A, (**B**) Group B, (**C**) Group C, and (**D**) Group D.

**Figure 8 materials-14-00913-f008:**
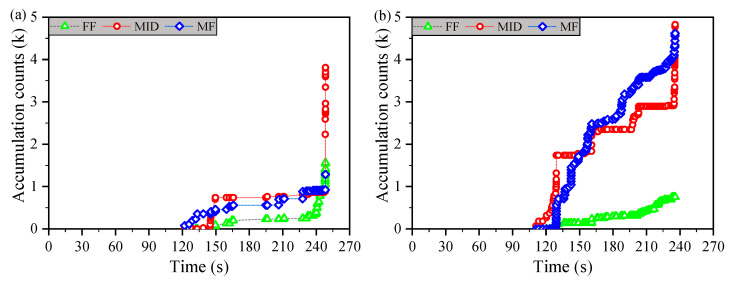
Acoustic emission damage mode of the accumulation counts–time relationship graphs: (**a**) Group A, (**b**) Group B, (**c**) Group C, and (**d**) Group D.

**Figure 9 materials-14-00913-f009:**
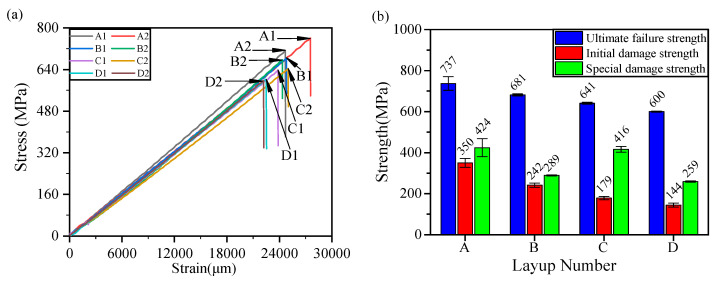
Ultimate failure strength, initial damage strength, and special damage strength ((**a**) stress vs. strain, (**b**) three kinds of strengths vs. layup number).

**Figure 10 materials-14-00913-f010:**
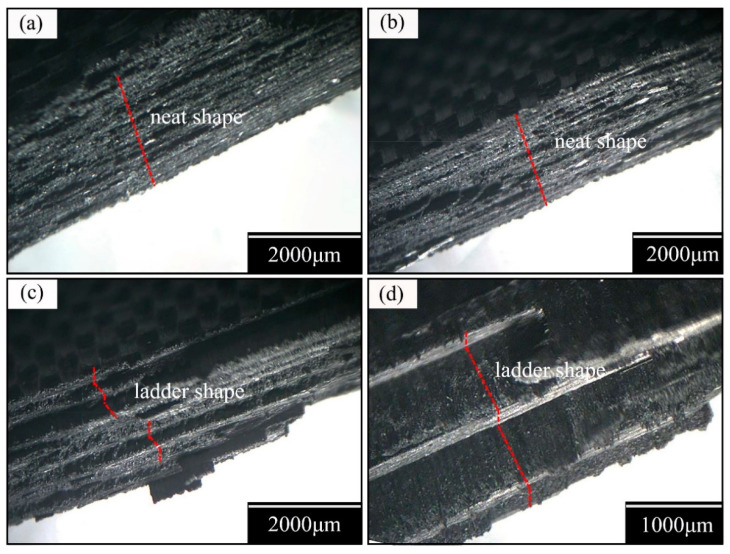
Fracture morphologies under an optical microscope: (**a**) Group A, (**b**) Group B, (**c**) Group C, and (**d**) Group D.

**Figure 11 materials-14-00913-f011:**
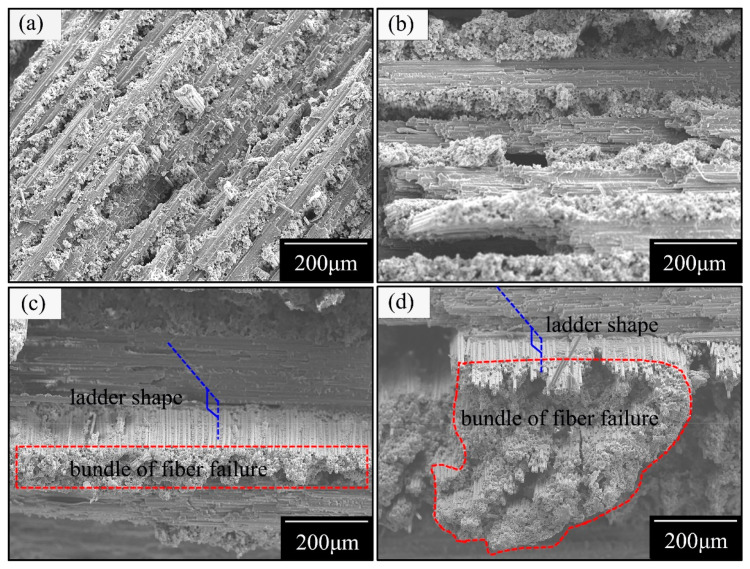
Fracture morphologies under SEM: (**a**) Group A, (**b**) Group B, (**c**) Group C, and (**d**) Group D.

**Table 1 materials-14-00913-t001:** Layer design and dimensions for tensile testing.

Groups	Layups	Thickness (mm)	Length (mm)	Width (mm)
A	0_2_[[90/0]_16s_]0_2_	2.6	250	25
B	0_2_ [[90_2_/0_2_]_8s_]0_2_	2.6	250	25
C	0_2_ [[90_4_/0_4_]_4s_]0_2_	2.6	250	25
D	0_2_ [[90_8_/0_8_]_2s_]0_2_	2.6	250	25
E	[90] _68_	2.6	250	25
F	[0] _68_	2.6	250	25

**Table 2 materials-14-00913-t002:** Parameter setting related to the acoustic emission (AE) equipment.

Channel Threshold [25]	Acquisition Frequency	Center Frequency	Acquisition Point
52 dB	2 MHz	255 kHz	2048

**Table 3 materials-14-00913-t003:** Specific frequencies corresponding to various damage modes.

Damage Modes	MID	MF	FF
Frequency range/kHz	0–85	240–260	261–304

## Data Availability

The data presented in this study are available on request from the corresponding author.

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
