# Peer review of "Mechanical Properties of Thin-Ply Composites Based on Acoustic Emission Technology"

_materials, 2021, doi:10.3390/ma14040913_

Round 1

Reviewer 1 Report

The article is well prepared and interesting, but it requires some improvements.

  1. Please describe more clearly the novelty of the study.
  2. When describing the test method (2.2.) and equations (1-3), it is necessary to add appropriate references. How did you determine the threshold level (Figure 3)? What criteria were used for this? Please adjust to each other the designations in Figure 3 and in formulas (1-3) to make it clearer for readers.  Please explain in the text what is t1, t2, ΔT, Δt and show this values in Figure 3.
  3. The characteristic frequencies of MID, MF, and FF were identified. What an how do these characteristic frequencies show, what conclusion can be drawn from this? Describe this in more detail.
  4. Describe in more detail the directions of the practical use of the research results and their limitations. Please provide a comparison of your research with the results of other authors. What are the differences, advantages and disadvantages of your approach?
  5. The conclusions look like a poor enumeration of what has been done. The conclusions require the addition of more specificity and analytics.

Author Response

Please see the attachment named "response letter I", thanks.

Reviewer 2 Report

Materials

Mechanical Properties of Thin-Ply Composites Based on Acoustic 2 Emission Technology

Review

The work concerns thin-film composites and the mechanisms of their destruction. The analysis concerns the sites of damage with respect to the matrix in terms of initial (MID), and final (MF) damage and fiber final(FF) damage, as well. The mechanism of demage in the composites was assessed based on the analysis of acoustic emission (AE) signals under tensile load in combination with the microstructural observations by light and electron scanning microscopy.

The work is interesting and will find many readers due to the topic undertaken and the presented results.

After completing the deficiencies and minor corrections, I recommend it for publication in the journal Materials

Work deficiencies and minor remarks

P.2, line 76 materials pre-pregs signature is not clear, please define the material type (by chemistry, phase composition…),type (and form) and impregnation (resign, method).

p.4 line 116. There is no information about samples preparation for microscopic observation.

Fig.4a add scale bar and number

Fig.6. Delete SE operation information bars, add scale bars and numbers in suitable font size. No EDS maps nor spectra to presented  images.

The text on photos is difficult to see, too low contrast is used, text (color and contrast) needs improvementFig.7C and D, accumulation energy axis, Please unify units (density)Fig. 10. Add scale bars and scale numbersFig.11 scale numbers are too small. Add some description to presented images (to underline the identified damage mode)

Author Response

Please see the attachment named "response letter II", thanks.

Round 2

Reviewer 1 Report

The answers are satisfactory. I think the article can be published.